Methods

# De novo prediction of cell-type complexity in single-cell RNA-seq and tumor microenvironments

Jun Woo[1,2], Boris J. Winterhoff[2,3], Timothy K. Starr[2,3], Constantin Aliferis[1], Jinhua Wang[1,2]

**Recent single-cell transcriptomic studies revealed new insights into cell-type heterogeneities in cellular microenvironments unavailable from bulk studies. A significant drawback of currently available algorithms is the need to use empirical parameters or rely on indirect quality measures to estimate the degree of complexity, i.e., the number of subgroups present in the sample. We fill this gap with a single-cell data analysis procedure allowing for unambiguous assessments of the depth of heterogeneity in subclonal compositions supported by data. Our approach combines nonnegative matrix factorization, which takes advantage of the sparse and nonnegative nature of single-cell RNA count data, with Bayesian model comparison enabling de novo prediction of the depth of heterogeneity. We show that the method predicts the correct number of subgroups using simulated data, primary blood mononuclear cell, and pancreatic cell data. We applied our approach to a collection of single-cell tumor samples and found two qualitatively distinct classes of cell-type heterogeneity in cancer microenvironments.**

## Introduction

Gene expression heterogeneities on the level of individual cells reflect key biological features not apparent from bulk properties, promising novel insights into molecular mechanisms underlying, e.g., development of neurons (Poulin et al, 2016), stem cell biology (Wen & Tang, 2016), and cancer (Navin, 2015; Winterhoff et al, 2017; Cieślik & Chinnaiyan, 2018; Nguyen et al, 2018). Recent advances in single-cell transcriptome profiling techniques using RNA-sequencing (RNA-seq; Ozsolak & Milos, 2011; Ziegenhain et al, 2017), together with customized computational methods (Buettner et al, 2015; Bacher & Kendziorski, 2016; Ilicic et al, 2016; Alpert et al, 2018; Edsgärd et al, 2018; Sinha et al, 2018; Soneson & Robinson, 2018; Kiselev et al, 2019), enabled significant progress in understanding such single-cell features (Tanay & Regev, 2017). Particularly noteworthy is the increased throughput of single-cell assays made possible by droplet-based barcoding technologies (Macosko et al, 2015), with cells in a typical sample numbering thousands or more (Zheng et al, 2017).

The ability to identify known cell types and discover novel cell groups is key to analyzing such data. Although classical unsupervised clustering and more recent dimensional reduction methods have been successfully adapted to single-cell RNA-seq data (Grün et al, 2015; Macosko et al, 2015; Bacher & Kendziorski, 2016; Li et al, 2017), a common drawback is the need to specify the degree of complexity in clustering, either by fixing the total number of subgroups anticipated or by choosing a resolution parameter controlling the extent of dimensional reduction. Because the degree of cell-type diversity expected from data is often unknown in real applications, a clustering approach capable of inferring the number of cell types present in a sample solely based on statistical evidence would provide a significant advantage, freeing cell-type classification and discovery process from potential resolution bias.

The question of how to determine the number of clusters in unsupervised clustering analysis has a long history in statistical literature (Milligan & Cooper, 1985; Tibshirani et al, 2001). Nevertheless, only a few currently available single-cell RNA-seq analysis pipelines provide such capability (Kiselev et al, 2019): SC3 uses principal component analysis (PCA) and compare eigenvalue distributions with that of random matrices to pick the most likely number of principal components (Kiselev et al, 2017); SINCERA (Guo et al, 2015) and RaceID (Grün et al, 2015) use statistics comparing intercluster versus intracluster separations; SNN-Cliq (Xu & Su, 2015) provides an estimate within a graph-based clustering approach. These existing choices thus either rely on indirect quality measures of multiple clustering solutions or significance tests associated with dimensional reduction.

In Bayesian formulation of general unsupervised clustering, in contrast, the number of clusters is just one of many hyperparameters, whose statistical support can rigorously be examined via Bayesian model comparison (Held & Ott, 2018): possible choices for the number of clusters can be compared quantitatively via marginal likelihood (or *evidence*, the probability of seeing data given a specific number of subgroups). In application point of view, a shift to Bayesian statistics therefore enables a comprehensive

---

[1]Institute for Health Informatics, University of Minnesota, Minneapolis, MN, USA   [2]Masonic Cancer Center, University of Minnesota, Minneapolis, MN, USA   [3]Department of Obstetrics, Gynecology and Women's Health, University of Minnesota, Minneapolis, MN, USA

Correspondence: wangjh@umn.edu

and powerful clustering approach, where clustering depth, assignment of individual cells into clusters, and characteristics of each cluster all emerge as collective analysis outcomes. To our knowledge, Bayesian model comparison is yet to be applied to single-cell RNA-seq analyses. Here, we developed and tested such a method for inferring and assessing the degree of heterogeneity in single-cell samples using Bayesian statistics and identifying the range of most appropriate number of clusters.

For the actual subgroup identification, we chose nonnegative matrix factorization (NMF) (Lee & Seung, 1999), an unsupervised machine-learning method of dimensional reduction, where a high-dimensional data matrix with nonnegative elements is factorized into a product of two matrices sharing a common, low dimension—the *rank* (Lee & Seung, 2000). Single-cell RNA count data are inherently nonnegative and typically sparse, making them ideal for NMF analysis. Earlier studies of bulk data and recent single-cell applications (Brunet et al, 2004; Carmona-Saez et al, 2006; Kim & Park, 2007; Puram et al, 2017; Zhu et al, 2017; Filbin et al, 2018; Ho et al, 2018) were all based on maximum likelihood (ML) formulation of the NMF algorithm (Gaujoux & Seoighe, 2010). The need to resort to quality measures of factorization (Brunet et al, 2004; Gaujoux & Seoighe, 2010) to choose its optimal value compromises the predictive power of ML-NMF, as with other clustering methods involving adjustable parameters controlling the degree of cell-type diversity. In contrast, we use NMF as one of possible dimensional reduction engines facilitating Bayesian model comparison and focus instead on the resulting capability to evaluate different choices of rank values. We adapted the variational Bayesian formulation of NMF (Cemgil, 2009) for barcoded single-cell RNA-seq data.

Cell-type heterogeneities in carcinoma samples pose a unique analytic challenge, with complex interplay of immune, stromal, and malignant epithelial cells playing key roles in the development and homeostasis of the tumor ecosystem (Li et al, 2016). Despite its predominance among cancer types, studies of single-cell transcriptomic heterogeneities in solid tumors are still in early stages (Jaskowiak et al, 2018). As a major application of our approach, we present analyses of available single-cell tumor samples, characterizing the range and depth of tumor microenvironment heterogeneities encountered in different cancer types.

# Results

## Optimal cell-type separation is determined by data

We implemented ML and Bayesian NMF (bNMF) algorithms for single-cell RNA count data (see the Materials and Methods section). Briefly, bNMF combines the NMF-based Poisson likelihood of RNA count data with gamma-distributed prior distributions for two-factor matrices (basis $W$ and coefficient $H$) (Cemgil, 2009) (Fig 1A). The mean counts are given by the matrix product $WH$, with inference optimizing both the factor matrices and hyperparameters of the priors simultaneously. The most likely rank is determined by comparing evidence (marginal likelihood of data conditional to hyperparameters $\Theta$ and rank $r$) for a range of rank values (Fig 1B):

$$\Theta^* = \arg\max_\Theta \Pr(X|\Theta, r), \tag{1}$$

$$r_{opt} = \arg\max_r \Pr(X|\Theta^*, r), \tag{2}$$

where $X$ is the RNA count data. We used the log evidence per matrix element, regarded as a function of rank, as the primary measure of statistical significance. Its difference between two rank values can then be related to Bayes factor (Kass & Raftery, 1995; Held & Ott, 2018): we used a conservative Bayes factor threshold 3 for statistically significant model differences in determining the optimal

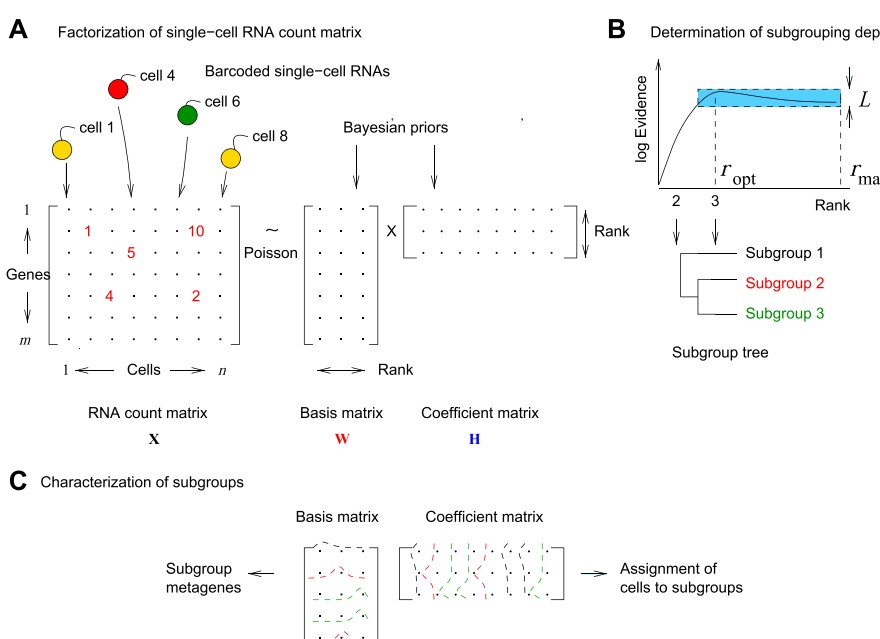

**A** Factorization of single−cell RNA count matrix

**B** Determination of subgrouping depth

**C** Characterization of subgroups

**Figure 1. bNMF for single-cell RNA-seq clustering.**
**(A)** RNA count matrix derived from droplet-based single-cell RNA-seq data is modeled as a Poisson realization of the mean given by a product of basis $W$ and coefficient $H$ matrices sharing a common dimension *rank*. Factorization infers these matrices for varying rank values using gamma priors. **(B)** We find the optimal rank maximizing log evidence or marginal likelihood of hyperparameters given the data. Heterogeneity class is determined by the shape of evidence profile: in type I, the difference in evidence between the maximum at rank $r_{opt}$ and the value at $r_{max}$ is larger than the threshold $L$; in type II, this difference is within $L$. The threshold is given by $L = (\ln T)/m$, where $T$ is the lower bound of Bayes factor for statistical significance. The factorization solutions for ranks from 2 to $r_{opt}$ are then used to construct the subgroup tree, which connects subgroups under successively increasing ranks. This tree provides a global view on the structure of cell-type heterogeneity on varying resolution. **(C)** Factor matrices $W$ and $H$ corresponding to the optimal rank are used to identify metagenes (genes distinguishing a given subgroup from the rest), characterize subgroups into known or novel cell types, and to assign individual cells into subgroups.

rank (Fig 1B and the Materials and Methods section). After factorization, the two-factor matrices yield metagene lists and subgroup membership of all cells (Fig 1C).

We first characterized the performance of bNMF using simulated data (Fig 2A–D). With data sets generated from $m$ = 100 features ("genes") and $r$ = 10 subgroups of 20 cells ($n$ = 200), we factorized the count data with varying rank $r$ using ML-NMF and computed two quality measures: dispersion and cophenetic correlation (the Materials and Methods section). Dispersion increased with increasing rank, saturating at $r \approx 10$ (Fig 2A). Cophenetic correlation (Brunet et al, 2004) showed a similar behavior with a maximum at $r$ = 10 (Fig 2B) and a narrow overall range of values close to 1.

We used bNMF to compute log evidence (Fig 2C), which increased linearly to reach a sharp maximum at rank 10. For higher rank values, log evidence decreased moderately. This trend remained unchanged for larger matrices up to sizes more typical of real data ($m$ = 2,000 and $n$ = 2,000; Fig 2C). In ML-NMF, likelihood is equal to the negative generalized Kullback–Leibler (KL) divergence, a distance measure distinct from Euclidean distance (see the Materials and Methods section). In bNMF, the generalized KL divergence is weighted by the prior distribution rather than minimized. As expected from this distinction, the Euclidean distance and generalized KL divergence both showed sharp cusps at rank 10 (Fig S1A and B), whereas for higher ranks, their magnitudes decreased weakly and remained similar for ML-NMF and bNMF, respectively. Thus, for these simulated data sets with 10 subgroups, ML-NMF predicted the correct rank well via two quality measures, and bNMF yielded a clear and unambiguous choice of the optimal rank. We also used a simulated data set of rank 5 to characterize how relative outlier cells in expression counts would be classified by bNMF factorization: the relative outliers identified by minimum covariance determinant method (Hubert & Debruyne, 2010) were predominantly located within the $t$-distributed Stochastic Neighbor Embedding (tSNE) (van der Maaten & Hinton, 2008) plot near the termini of branches separately forming individual subgroups (Fig S2), suggesting that bNMF would be resistant to overclustering of moderate outliers. As a representative choice from existing methods relying on specification of parameter(s) controlling clustering depth, we applied Seurat (Macosko et al, 2015) to the same simulated data with a range of resolution parameter values. With increasing resolution, the number of subgroups obtained showed consecutive jumps to reach 10 (Fig 2D).

We further tested the convergence of bNMF inference using a different simulation scheme, where factor matrices $W$ and $H$ were generated from $\gamma$ priors with known hyperparameters (Fig S3). With increasing sample size, the evidence profile converged to a shape as in Fig 2C and the predicted optimal rank and hyperparameters became more sharply peaked around the correct values.

We next compared these algorithms using the fresh PBMC single-cell data set (Zheng et al, 2017; Fig S4A and Table S1). To test the dependence of the number of subgroups on sample sizes, we used two different subsamples ($n$ = 34,289 and $n$ = 6,857) derived from the full data. We first characterized evidence profiles with the smaller data set under ML-NMF, bNMF, and PCA (Fig 2E–H). Both dispersion and cophenetic correlation from ML-NMF were maximal near $r$ = 2; dispersion increased moderately for large $r$, whereas cophenetic correlation remained low for $r$ > 10 (Fig 2E and F). The log evidence

from bNMF exhibited a sharp increase with increasing rank for $2 \leq r \leq 6$ and decreased slightly for larger ranks. The rank with maximum evidence was $r$ = 9. Seurat led to a monotonic increase in the number of subgroups with increasing resolution from $r$ = 5 to $r$ = 21 (Fig 2H). In contrast, both Euclidean distance and KL divergence decreased monotonically with increasing ranks under ML-NMF and bNMF (Fig S1D and E). The bNMF evidence profile was robust against varying sample sizes, reaching maximum at $r \approx 6$ and remaining similar or decreasing slightly for larger ranks (Fig 2G).

We further compared bNMF rank profiles with the numbers of clusters predicted by existing algorithms for six small single-cell data sets (Yan et al, 2013; Biase et al, 2014; Deng et al, 2014; Pollen et al, 2014; Kolodziejczyk et al, 2015; Goolam et al, 2016) with well-known cell-type complexity (e.g., embryonic stem cells in early development): "gold standard" data sets used in published works assessing SC3 (Kiselev et al, 2017) and SIMLR (Wang et al, 2017). In many cases (Fig 2I, K, M, and N), the number of cell types expected from experimental design coincided with the lowest rank regions where bNMF-derived evidence profile became relatively flat. At the same time, apparent overestimations of the number of clusters by other methods often fell within such flat regions (Fig 2I, J, L, M, and N), providing a possible explanation for the lack of consensus among different methodologies: many data sets exhibit evidence profiles that are monotonically increasing up to a certain rank, beyond which statistical support remains similar.

In summary, although all three algorithms performed reasonably well for simulated data sets with simple compositions, NMF provided a means to assess the subtype complexity without the need to set adjustable parameters (Fig 2A–D). The bNMF enabled a statistically well-controlled comparison via the evidence profile, which unambiguously predicted the number of subgroups supported by PBMC data (Fig 2E–H). Derivation of evidence profiles for benchmark single-cell data sets demonstrated that bNMF reveals a much more comprehensive picture of how statistical support varies with the number of clusters than in existing computational methods estimating a single clustering depth (Fig 2I–N).

## bNMF infers depth of heterogeneity in PBMC/pancreatic cells

We next characterized bNMF cell-type separation outcome of the PBMC ($n$ = 34,289) using the metagenes from basis matrix $W$ (Fig 1C) under rank 9 (Fig 2G). Most of the top metagenes clearly distinguished each subgroup from the rest, whereas a small proportion of them featured in more than one subgroups (Fig 3A). We used correlations between the mean expression counts of subgroups and those of purified blood cell types (Zheng et al, 2017), along with metagene and markers (Foell et al, 2007; Walzer et al, 2007; Kallies, 2008; Quann et al, 2011; Lu et al, 2017) to annotate major components of nine clusters (Fig 3A and B).

The bNMF inference results from rank 2 to 9 provide cell-type separation outcomes with increasing resolution up to the optimal rank, beyond which statistical support from data no longer improves. Using cluster membership of all cells under these ranks, we constructed a hierarchical tree relating these subgroups (Fig 3B). The two subgroups at rank 2 separated cells into two branches, one containing B cells, NK cells, and monocytes and the other

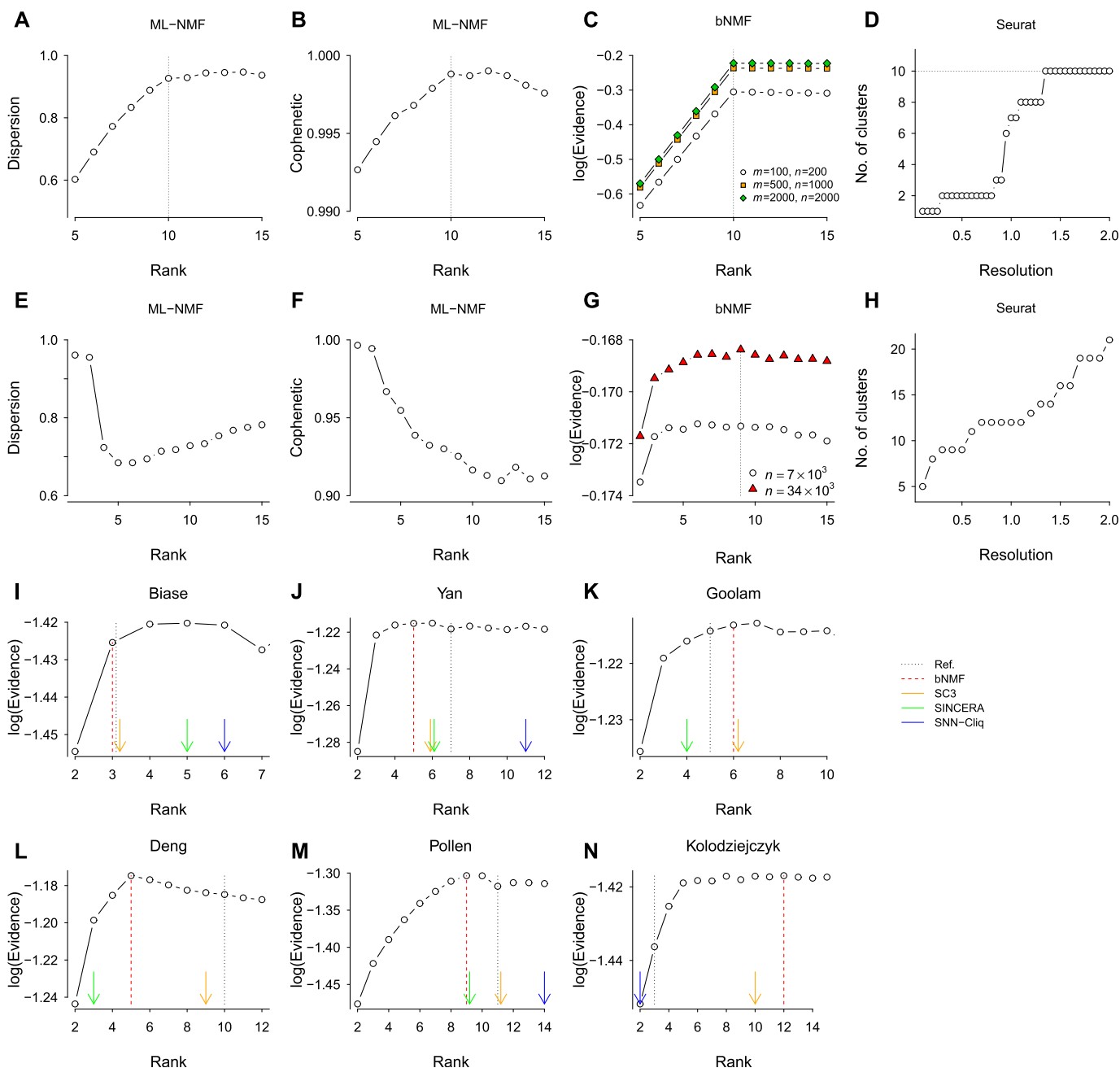

**Figure 2. Comparison of optimal rank determination by NMF (ML-NMF and bNMF) and other clustering methods.**
**(A–D)** Simulated data of 100 genes and 10 subgroups of cells (20 in each subgroup; 200 in total, except noted otherwise in (C)). ML-NMF narrows down the rank into an optimal range based on two quality measures, dispersion and cophenetic coefficient. **(C)** bNMF finds the correct rank 10 maximizing evidence. **(D)** Seurat (Macosko et al, 2015) requires specification of resolution parameter; the correct number of subgroups is reached as the upper bound with respect to resolution. **(E, F)** ML-NMF applied to PBMC single-cell data (Zheng et al, 2017). **(G)** bNMF applied to PBMC data sets of different sizes led to the optimal rank maximizing evidence as $r_{opt} \approx 9$. **(H)** PCA applied to PBMC yielded a wide range of subgroup numbers depending on resolution. **(I–N)** bNMF rank profiles and the number of clusters predicted by other computational algorithms applied to six gold standard data sets (Yan et al, 2013; Biase et al, 2014; Deng et al, 2014; Pollen et al, 2014; Kolodziejczyk et al, 2015; Goolam et al, 2016). The SC3 (Kiselev et al, 2017), SINCERA (Guo et al, 2015), and SNN-Cliq (Xu & Su, 2015) predictions are from Kiselev et al, 2017. The black dotted and red dashed lines are the number of major cell types expected from experimental design and the optimal rank from bNMF protocol, respectively. In (I), the total number of cells was small ($n = 49$) so that a large subset of factorization results in $W$ matrices had uniform columns for $r \geq 4$, implying $r_{opt} = 3$.

containing T cells. Intermediate levels of subgrouping within the tree revealed sub-branches linking B cells and monocytes, and naive/helper/regulatory versus effector/memory T cells. This global tree view under varying rank values facilitates biological interpretation of subgroups within the framework of NMF-enabled dimensional reduction. We applied t-SNE to the coefficient matrix $H$ elements and visualized the seven subgroups (Fig 3C). The proximity of subgroups within the map closely reflected their hierarchical

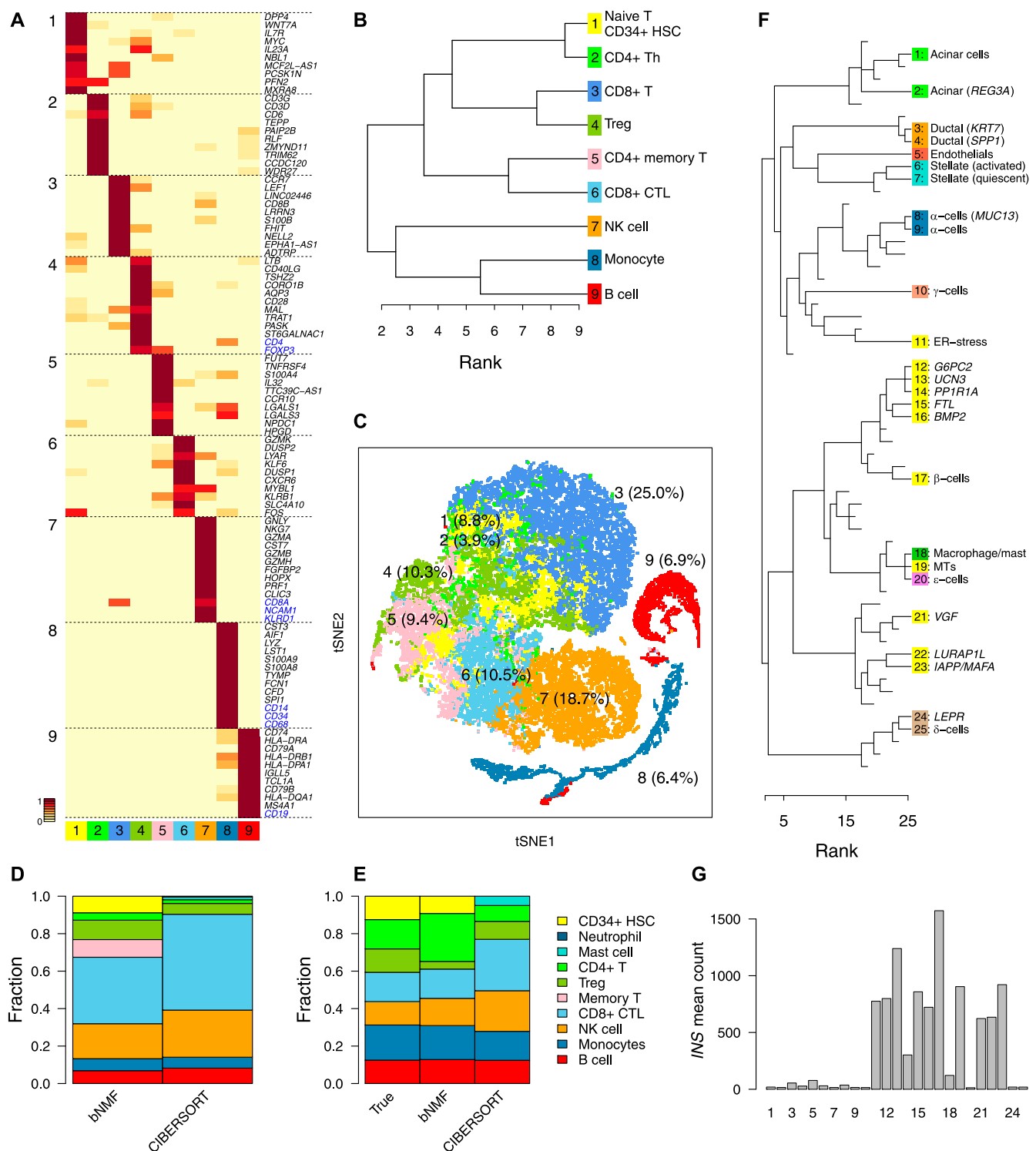

**Figure 3. bNMF subgrouping results for PBMC and pancreas data sets.**
**(A–C)** Results for the PBMC data set (*n* = 34,289). **(A)** Metagenes for subgroups derived from the factor matrix *W* under optimal rank 9 (Fig 2G). Heat map shows the relative magnitudes of matrix element $W_{ik}$ for each gene *i* and subgroup *k*, rescaled such that in each row, minimum and maximum correspond to 0 and 1. Up to 10 metagenes in addition to preselected markers per subgroup are shown. **(B)** Subgroup tree showing hierarchical relationships between subgroups under varying ranks from the lowest (2) to the optimal (9). Branching of a subgroup under a given rank into two under a successively larger rank was inferred by applying the majority rule (see the Materials and Methods section). **(C)** Visualization of subgroups with tSNE. Subgroup ID and composition of cells are indicated. **(D, E)** Comparison of cell type compositions predicted by bNMF and bulk data deconvolution method, CIBERSORT (Newman et al, 2015). Outcomes for the full fresh PBMC data and an example mixture of seven purified cell types are shown in (D) and (E), respectively. **(F)** Subgrouping of human pancreas cell data (Baron et al, 2016). Colors indicate major cell types. Insulin-producing *β*-cells are in yellow (see Figs S5 and S6 and Table S2). MT, metallothionein. **(G)** Mean RNA count of *INS* gene in each pancreas subgroup.

relationships (the two branches under rank 2 in Fig 3B roughly occupy the upper-left and lower-right portions in Fig 3C).

We further tested the capability of bNMF to identify biologically well-characterized cell types by analyzing human pancreatic single-cell data (Baron et al, 2016). The evidence profile indicated a range of optimal ranks of $r = 20 \sim 30$ and remained similar for higher ranks (Fig S5A). We used the metagene list under $r = 25$ (Fig S6) to identify all major cell types (Fig 3F), which included two acinar cell subgroups (1 and 2), the latter expressing *REG3A*, two $\alpha$-cell subgroups (8 and 9), $\gamma$-cells (subgroup 10), $\delta$-cells (subgroup 24 and 25), $\varepsilon$-cells (subgroup 20), ductal cells (subgroup 3 and 4), endothelials (subgroup 5), and stellate cells (subgroups 6 and 7, activated and quiescent, respectively). Although *INS* featured most strongly in subgroup 17, the distribution of insulin expression (Fig 3G) indicated that subgroups 11–16, 19, 21–23 also comprised $\beta$-cells. These subdivisions were further supported by their proximity in tSNE plots (Fig S5B–D). Notably, metagenes of subgroup 11 (*HSPA5* and *DDIT3*) linked it to proliferative $\beta$-cells with endoplasmic reticulum stress (Baron et al, 2016). We found one subgroup (subgroup 18) of (largely macrophage) immune cells. Our subgroup assignment was highly concordant with the cell type annotation by Baron et al (2016) (Table S2).

### bNMF classifies known cell types with high accuracy

We next tested the robustness of bNMF clustering applied to real data using mixtures of count data derived from purified PBMCs (Zheng et al, 2017). We generated multiple realizations of PBMC data sets of known composition by sampling fixed numbers of up to seven cell types—CD8$^+$ CTLs, B cells, monocytes, CD4$^+$ Th, regulatory T cells (Treg), NK, and hematopoietic stem cells (HSCs)—of equal proportions and performed bNMF inference for each realization. The distribution of optimal ranks gradually shifted to higher ranks as mixtures became more complex (Fig 4A–F). It was notable that the degree of shifts with the successive addition of new cell types reflected the novelty in the added cell type: the addition of Tregs and NK cells to mixtures already containing Th cells and CTLs (Fig 4C–E) led to only moderate shifts in optimal ranks to higher values, whereas the addition of HSCs led to a more substantial jump (Fig 4E and F). Typical shapes of evidence profiles showed two distinct qualitative trends: for mixtures with low complexity, there was a sharp and pronounced rank value with maximum evidence (Fig 4G) and statistical support decreased for larger rank values (type I). For complex mixtures, on the other hand, the evidence profile became relatively flat, with support for broader range of rank values above a threshold (Fig 4H; type II).

We quantified the reliability of subgroup assignment by the following procedure: we first determined the cell-type identities of subgroups obtained under rank 4 inferred for four-sample mixtures (Fig 4C) using metagenes. We then assigned cells into four subgroups using $H$ matrix elements and calculated classification score as the proportion of correctly classified cells. We obtained a mean score of 0.82 ± 0.08 (SD; Fig 4I). To further test identification of rare cell types, we used mixtures containing four cell types of which two had cell counts of ~10% of the rest, obtaining the score of 0.73 ± 0.08. Together, these tests indicated that bNMF enabled robust determination of optimal subgrouping depths and reliable assignment of individual cells into subgroups.

We further compared the cell-type identification of bNMF with that of a deconvolution procedure, where reference panels of expression patterns are used to infer cell-type compositions from bulk data (Avila Cobos et al, 2018). We used CIBERSORT (Newman et al, 2015) to estimate the proportion of cell types from RNA counts averaged over fresh PBMC cells and found a reasonable agreement with noticeable differences when compared with single-cell results (Fig 3D). We further characterized differences in cell-type proportion estimates from single-cell and deconvolution methods with a mixture of seven purified blood cells: the bNMF prediction (Fig 3E), where the major discrepancy arose in discriminating Treg from Th cells (also see Fig S7), was substantially closer to true proportions (Fig 3E), demonstrating the advantage of explicit single-cell data analysis compared with bulk deconvolution.

Because our algorithm takes cell-count matrix as input, it can be combined with improved quality control or preprocessing steps alleviating challenges in single-cell capture and counting protocols. Such challenges include the overabundance of zero counts thought to originate from incomplete sampling of low-number RNA molecules in individual cells (Lin et al, 2017; Li & Li, 2018). To demonstrate such a combined usage, we processed the cell-count matrix of one of the PBMC seven-cell-type mixtures in Fig 4F with scImpute (Li & Li, 2018). Imputation did not change the evidence profile (Fig S7A), where the optimal rank was 6 with rank 7 slightly lower but close in evidence value. The bNMF factorization results of the original and imputed count matrices (Figs S7B and 7C) showed that CD4$^+$ Th and Treg cells were clustered together in both cases, explaining the optimal rank of 6. Imputation enhanced the quality of cell-type resolution separating Th/Treg and CTL subgroups, resulting in a closer agreement of overall cell counts in each cluster in comparison to true cell counts (Fig S7D).

### Solid tumor cell cultures have limited heterogeneity

We next applied our algorithm to melanoma cell culture single-cell data (Gerber et al, 2017), which contain transcriptomes of tumor cells derived from three patients: two replicates of wild-type (WT), *BRAF* mutant-*NRAS* WT, and *BRAF* WT-*NRAS* mutant samples. The evidence profile of this in vitro data set (Fig 5A) showed a pronounced maximum near $r \sim 7$, decreasing sharply for higher rank values. This behavior was analogous to those for low complexity mixtures of immune cells (Fig 4G; type I). The tSNE visualization of seven subgroups closely reflected the patient of origin and mutation status (Fig 5B and C): the subgroups of cells from WT patient (subgroups 1–4) formed one major branch (Fig 5D), which included subgroups expressing oxidative phosphorylation and other melanoma-specific marker genes (Gerber et al, 2017) (subgroup 1), a highly proliferative subgroup expressing cell cycle and DNA repair genes (subgroup 2), and a stromal subgroup (subgroup 3; Fig 5E). The BRAF-mutant cells (subgroups 5–6) showed *CD34*, *BRAF*, and apoptosis-related genes as metagenes/markers, whereas *NRAS*-mutant cells had *NRAS* as a marker. Overall, this outcome was consistent with the expected low depth of heterogeneity in cultured tumor samples.

## Tumor microenvironments in vivo show two distinct classes of heterogeneity

We characterized the degree of cell-type heterogeneity in tumor microenvironments in vivo with six additional solid tumor data sets (Table S1 and Fig 6). Lavin et al (2017) studied the landscape of innate immune cells infiltrating lung adenocarcinoma. We obtained a rank profile with a relatively narrow range of optimal ranks (Fig 6A). The subgroups derived consisted of B cells, mast cells, NK cells, dendritic cells, monocytes, and tumor-/normal cell-associated macrophages (Fig S8). We also analyzed two glioma samples (oligodendroglioma [Tirosh et al, 2016b] and astrocytoma [Venteicher et al, 2017]), which both exhibited rank profiles (Fig 6B and C) similar to lung cancer immune cell results: together, these samples were characterized by an intermediate level of heterogeneity with

optimal rank of $r \sim 20$ and decreasing statistical support for higher ranks (type I; Fig 6A–C).

In contrast, the evidence profiles for three additional data sets—melanoma (Tirosh et al, 2016a, Fig 6D), immune cells in breast cancer (Azizi et al, 2018, Fig 6E), and head and neck squamous cell carcinoma (HNSCC; Puram et al, 2017, Fig 6F)—showed a different behavior, where evidence increased monotonically to reach a maximal level and remained similar for higher ranks (type II). We classified evidence profiles into these two classes unambiguously by comparing maximum evidence and evidence at maximum rank using a Bayes factor threshold (Fig 1B): although clear maxima existed in type I data sets (Fig 6A–C), global maxima were located at the highest rank considered in type II (Fig 6D–F). In type II data, the lowest rank with the evidence value within the threshold around the maximal level provides the most parsimonious description.

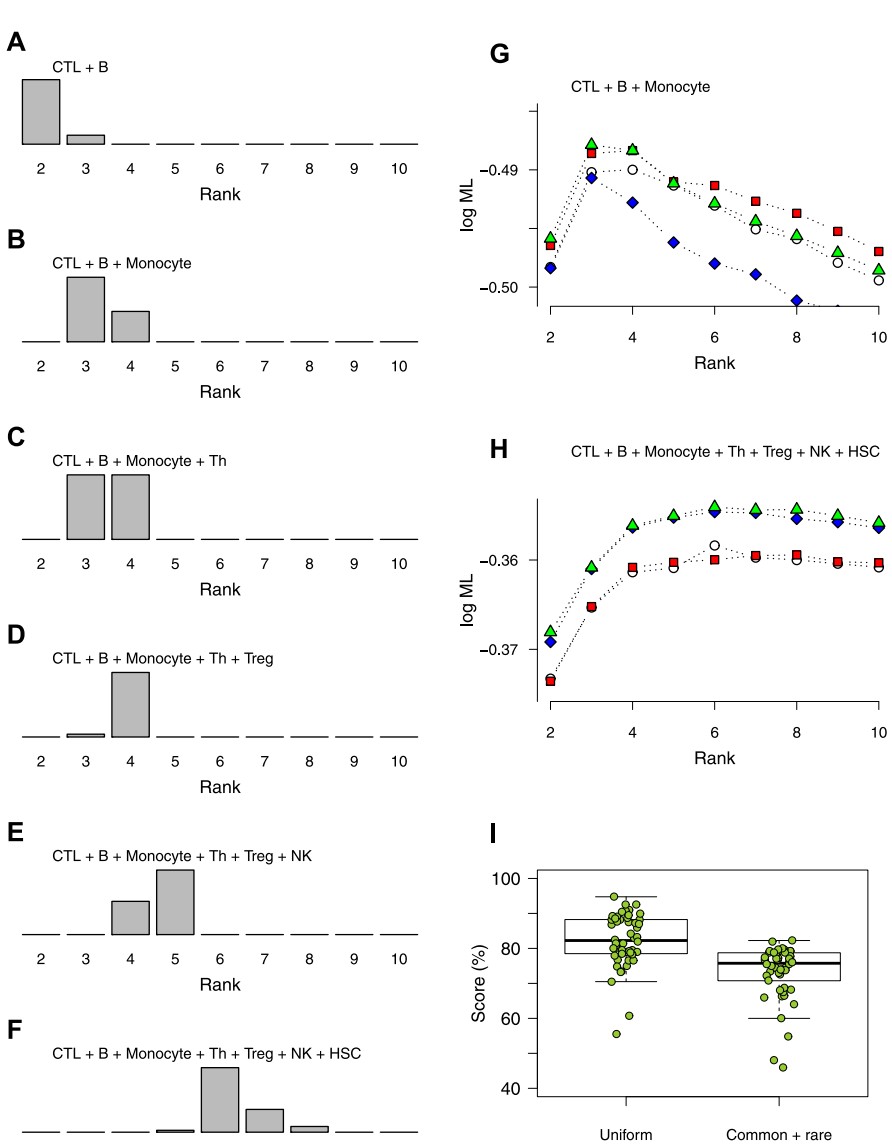

**Figure 4. Distributions of optimal ranks from bNMF inference applied to randomly sampled mixtures of purified blood cells.**
**(A–F)** Mixtures containing selections of CD8⁺ T cells (CTL), B cells (B), monocytes, CD4⁺ T cells (Th), regulatory T cells (Treg), NK cells (NK), and CD34⁺ HSCs, of varying compositions as indicated. **(G, H)** Examples of rank versus evidence profiles for mixtures of three (G) and seven (H) blood cell types. **(I)** Subgroup assignment scores (fraction of correctly assigned cells) of bNMF-based inferences applied to mixtures of four purified blood cell types shown in (C). Two sets of mixtures with different compositions were sampled, one with uniform cell counts ("uniform") and the other where three cell types were ~10% in count than the rest ("common + rare"). Mean scores are 0.82 (0.08, SD) and 0.73 (0.08) for uniform and common + rare cases, respectively.

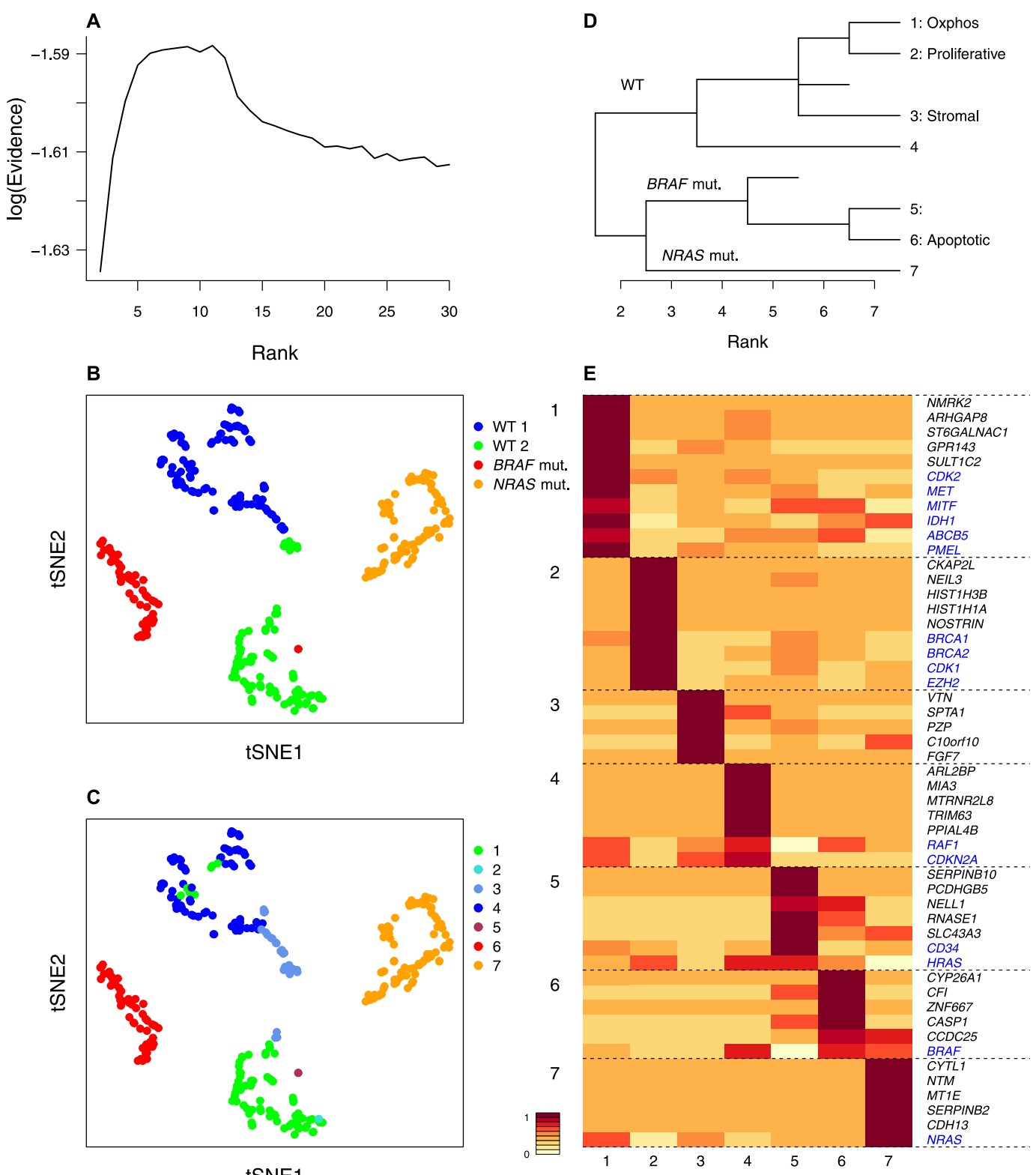

**Figure 5. bNMF subgrouping results of melanoma cell culture transcriptome data (Gerber et al, 2017).**
**(A)** Rank versus evidence profile. **(B, C)** tSNE visualizations of cells using bNMF *H* matrix elements colored by sample of origin in (B) and subgroup identity under rank 7 in (C). **(D)** Custer tree from rank 2 to 7. Oxphos, oxidative phosphorylation (Gerber et al, 2017). **(E)** Metagene map showing top five metagenes in each subgroup and marker genes (blue). mut., mutant.

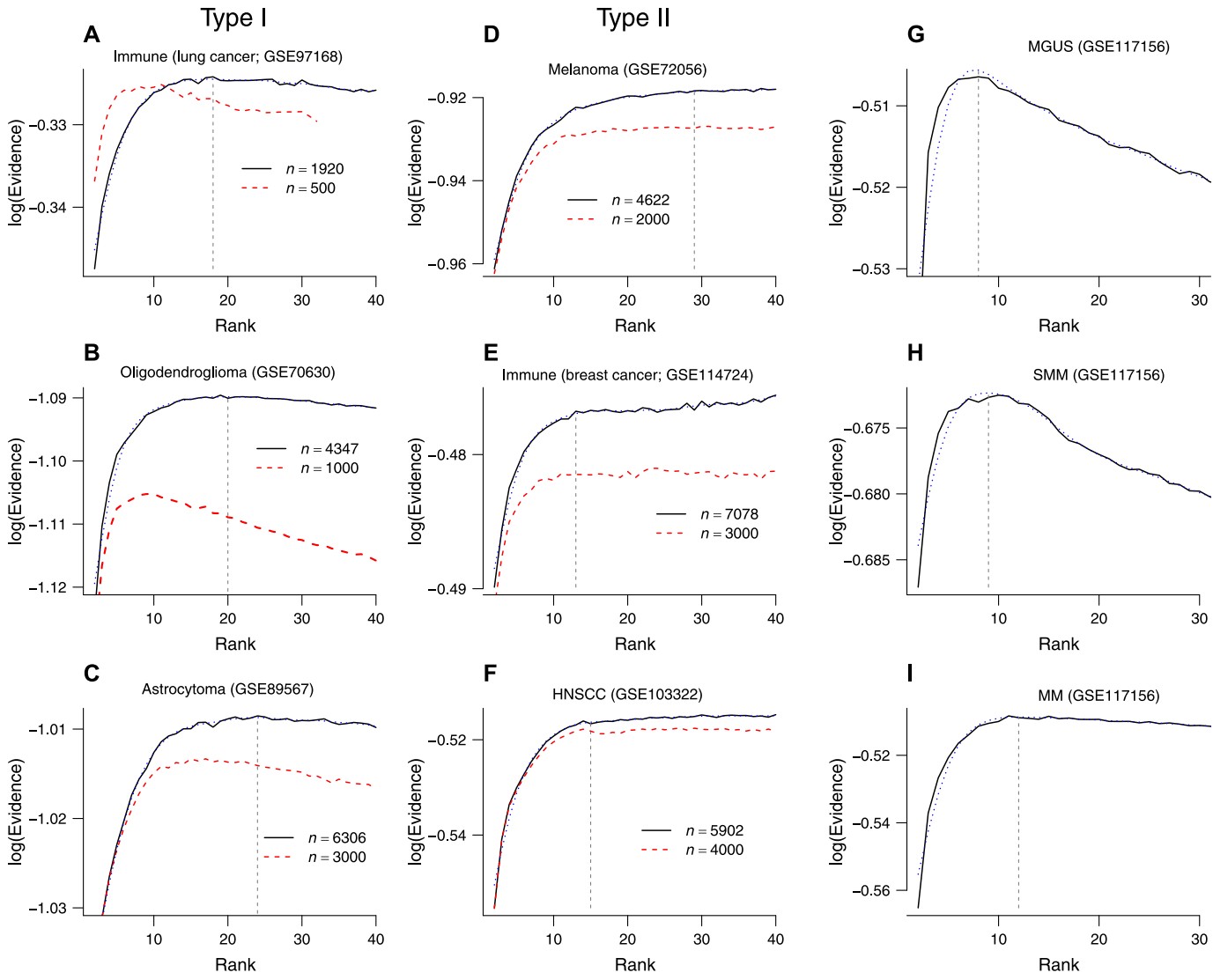

**Figure 6. bNMF clustering outcomes for in vivo solid tumor and myeloma samples.**
**(A–F)** Rank versus evidence profiles of data sets with accession numbers as indicated (Tirosh et al, 2016a, 2016b; Lavin et al, 2017; Puram et al, 2017; Venteicher et al, 2017; Azizi et al, 2018). Dotted blue lines are smooth-spline fits to data. Vertical dashed lines are locations of optimal rank. **(G–I)** Evidence profiles of MM samples in MGUS, SMM, and full MM stages (Ledergor et al, 2018).

To ensure that our classification did not depend on quality of statistics afforded by each data set, we repeated each inference after down-sampling, where sample sizes were reduced by a factor of 2 ~ 4. All three cases in type I retained their shapes with the locations of maxima shifted to lower ranks (Fig 6A–C, red dashed lines), suggesting that the pronounced maxima in evidence profiles observed for full data sets were statistically significant. In contrast, upon down-sampling, all three type II samples retained their shapes of asymptotic monotonicity with similar locations of optimal rank (Fig 6D–F). We additionally examined ML-NMF quality measures of two representative tumor samples, each from type I and II classes (oligodendroglioma and breast cancer immune cells; Fig S9). The rank-dependence of dispersion and cophenetic coefficients were qualitatively similar to those of PBMC (Fig 2E and F), with maxima at rank ~2, minima below rank ~20, and monotonic increases under large rank values (Fig S9).

We further characterized the composition of HNSCC sample, which contains primary and lymph node metastatic tumors from 18 patients (Puram et al, 2017). The subgroup tree (Fig S10A) showed a division at $r = 2$ into epithelial (subgroups 1–8) and immune/ stromal branches (subgroups 9–15). Major cell type assignments from bNMF were highly concordant with annotations by Puram et al (2017) (Fig S10B–D).

Given the fundamental roles somatic mutations play in cell-type heterogeneity of tumors, we reasoned that the type II–like behavior of high-complexity cancer microenvironments would be associated with relatively large degrees of somatic mutations. We explored such a connection between transcriptomic and DNA-level complexities using single-cell data sets from multiple myeloma (MM) patients (Ledergor et al, 2018): we characterized three sets of malignant plasma cell samples derived from patients at different

stages of disease progression: an asymptomatic, monoclonal gammopathy of undetermined significance (MGUS), a more advanced, smoldering multiple myeloma (SMM), and full MM stages. These disease stages exhibit progressively larger degrees of somatic copy number aberrations (Ledergor et al, 2018). The MGUS sample showed a clear type I behavior with the optimal rank of 8 and a strong monotonic decrease in evidence for higher ranks (Fig 6G). The SMM sample showed a broader peak at rank 9 (Fig 6H). The MM sample result, in contrast, was strongly indicative of a borderline behavior where type I would transition into type II (Fig 6I). This progression of evidence profiles supports the view that cancer disease progression and increases in somatic mutation load would typically cause a gradual replacement of type I by type II behaviors.

# Discussion

Our approach for single-cell RNA-seq analysis confers a unique capability of assessing the degree of cell-type heterogeneity via unsupervised clustering with the number of subgroups rigorously determined from data. We showed with simulated data sets and existing PBMC/pancreatic single-cell data that the appropriate depth of subgrouping is generally dictated by data at hand and is largely independent of sample sizes. Our method allows us to not only infer this degree of complexity but also identity cellular subtypes with high accuracy and consistency (Figs 2, 3, and 4). In particular, the high degree of heterogeneity we found among pancreatic $\beta$-cells (Fig 3F and G) is consistent with existing experimental evidences (Wang & Kaestner, 2018).

The prominence of peaks signifying the optimal rank—the range of heterogeneity most appropriate for the data set at hand—in samples of relatively low complexity (e.g., Figs 4G, 5A, and 6A–C), where statistical support clearly decreases for larger ranks, illustrates a key difference between ML approaches and bNMF: in ML methods, larger ranks using more parameters would generally result in better fit unless penalized. In contrast, explicit priors used in bNMF ($\gamma$ distribution in our case) prevent overfitting.

Our characterization of solid tumor microenvironments highlights the diversity in the degree of heterogeneity and the importance of assessing it adequately in transcriptomic studies. The highly pronounced and low value of optimal rank observed for in vitro tumor cell culture (Fig 5A) is in contrast with in vivo tumor microenvironments, which showed intermediate (type I, Fig 6A–C) to high (type II, Fig 6D–F) levels of heterogeneity. The latter two classes of heterogeneity each showed a relatively clear optimal rank and a lower bound for subgroup number with evidence equally supporting all higher depths, respectively. Although two type II samples (melanoma and HNSCC) contained primary and metastatic tumors from multiple patients (Table S1), which presumably contribute to heterogeneity, the multiplicity of patient/tumor of origin comprising each data set did not determine heterogeneity class by itself: the breast cancer immune cell data derived from a single patient belonged to type II (Fig 6E), whereas two type I cases (gliomas, Fig 6B and C) contained 6 and 10 patients, respectively.

The tumor types and their heterogeneity classes in Fig 6B–F instead are broadly consistent with their known relative somatic mutation loads (glioma < breast cancer < HNSCC < melanoma;

Alexandrov et al, 2013). A type II behavior in tumor samples thus suggests extensive cell-type heterogeneities spanning a substantial range of resolution, possibly down to levels reaching individual cells. Such a complex gene expression signature spanning multiple levels could arise from extensive diversification of tumor cells through somatic mutation, as suggested by the progression of MM samples in Fig 6G–I. In contrast, a single or narrow range of optimal ranks would signify a well-defined, finite set of subgroups, with cells in each subgroup relatively homogeneous in their expression profiles.

Although we adopted the "pooled" analysis approach for samples containing multiple tumors, one may instead seek to extract shared molecular-level profiles independent of patient or tissue of origin, which would require incorporation of a batch effect-removal strategy (Dal Molin & Di Camillo, 2018; Haghverdi et al, 2018). Such multi-sample extension may take the form of a statistical procedure deriving a consensus subgrouping depth among multiple values optimal for each constituent sample.

# Materials and Methods

## ML-NMF

We implemented ML (Lee & Seung, 2000) and variational bNMF inference with $\gamma$ priors (Cemgil, 2009) for factorization of count data. A statistical inference-based formulation of NMF regards each element of count matrix $X$ ($m$ rows for gene and $n$ columns for cells) as a realization of the sum of $r$ Poisson random variables, $X_{ij} = \sum_{k=1}^{r} S_{ikj}$, where $S_{ikj} \sim \text{Poisson}(\lambda = W_{ik}H_{kj})$ is a "latent source" variable. The matrices $W$ and $H$ are the basis and coefficient factor matrices, each of dimension $m \times r$ and $r \times n$, respectively. The intermediate dimension $r$ (rank) typically satisfies $r \ll m$ and $r \ll n$.

Using the known property that the distribution of a sum of Poisson random variables is Poisson with mean equal to the sum of individual means, one has:

$$X_{ij} \sim \text{Poisson}\left[\lambda = \sum_{k} W_{ik}H_{kj} = (WH)_{ij} \equiv \Lambda_{ij}\right] \qquad (3)$$

One can then write for the likelihood of data,

$$\ln \Pr(X|W,H) = \sum_{ij} \ln\left[\Pr(X_{ij}|W,H)\right] = \sum_{ij} \ln\left[e^{-\Lambda_{ij}} \Lambda_{ij}^{X_{ij}} \middle/ X_{ij}!\right]$$
$$= \sum_{ij}\left(X_{ij} \ln \Lambda_{ij} - \Lambda_{ij} - X_{ij} \ln X_{ij} + X_{ij}\right), \qquad (4)$$

where Sterling's approximation was used in the second line. The likelihood then takes the form of:

$$\ln \Pr(X|W,H) = \sum_{ij}\left[X_{ij} \ln \frac{(WH)_{ij}}{X_{ij}} + X_{ij} - (WH)_{ij}\right]. \qquad (5)$$

The right-hand-side of Equation (5) is the negative of generalized KL divergence (Lee & Seung, 2000), which is minimized upon ML condition. An expectation–maximization treatment applied to Equation (5) (Cemgil, 2009) leads to the iterative update rule for $W$ and $H$ first derived by Lee & Seung (1999).

We used ML inference with randomized initial conditions, where multiple iterations were seeded by identically distributed initial matrix elements. Convergence was tested with fractional changes to log likelihood below a cutoff ($10^{-5}$). Quality measures we considered were dispersion and cophenetic correlation. The dispersion was defined with respect to the consistency matrix. Consistency matrix $C$ is an $n \times n$ matrix with elements $C_{jl} = E(\delta_{jl})$, where $\delta_{jl}$ is the Kronecker $\delta$ equal to 1 if cell $j$ and cell $l$ belong to the same cluster and zero otherwise, and the mean is taken over factorization results with different initial conditions. A given cell $j$ is assigned to the cluster $r$ within a factorization outcome, where $r = \arg\max_k H_{kj}$. The dispersion, a measure between 0 and 1 for the consistency of cluster assignment over multiple inferences, was defined as:

$$D = \frac{4}{n^2} \sum_{jl} \left( C_{jl} - \frac{1}{2} \right)^2 = \frac{1}{n} + \frac{8}{n^2} \sum_{j<l} \left( C_{jl} - \frac{1}{2} \right)^2, \quad (6)$$

i.e., the mean deviation of the consistency matrix from the null value 1/2. The factor of 4 rescales the value such that $\max(D) = 1$, and in the second expression, we separated the diagonal term for which $C_{jj} = 1$; the second summation is over the upper triangular part of $C$. Cophenetic correlation was defined as:

$$P = \text{cor}\left( 1 - C_{jl}, h_{jl} \right), \quad (7)$$

i.e., the correlation between consistency matrix and the height $h_{jl}$ within the dendrogram from hierarchical clustering at which cell $j$ and cell $l$ merge (Sokal & Rohlf, 1962; Brunet et al, 2004). The cophenetic correlation $P$ measures the degree to which dissimilarity between two cells $1 - C_{jl}$ is preserved in hierarchical clustering. We used the "hclust" function in R with "average" method for the computation of $P$.

### bNMF

We used Bayesian inference, evaluating the marginal likelihood or evidence,

$$\Pr(X|\Theta, r) = \int dW dH \sum_S \Pr(X|S)\Pr(S|W, H)\Pr(W, H|\Theta, r), \quad (8)$$

where $\Theta$ is the set of hyperparameters for the prior distribution of factor matrices $W$ and $H$. Both hyperparameters and rank $r$ can be chosen by maximizing evidence [Equations (1) and (2)]. In practice, hyperparameters are updated during iteration for a given rank and the inference is repeated for multiple rank values. The resulting (log) evidence values can then be compared to find $r_{opt}$. We assumed all matrix elements were identically distributed by $\gamma$ priors with shape $\alpha$ and rate $\beta$ parameters:

$$\Pr(W, H|\Theta, r) = \prod_{i,k} \text{Gamma}\left( W_{ik} \middle| \alpha = a_w, \beta = \frac{a_w}{b_w} \right) \prod_{k,j} \text{Gamma}$$
$$\left( H_{kj} \middle| \alpha = a_h, \beta = \frac{a_h}{b_h} \right), \quad (9)$$

such that $\Theta = \{a_w, b_w, a_h, b_h\}$ We used update equations for the posterior mean of latent and factor elements resulting from a variational approximation to Equation (8) (Cemgil, 2009). We typically held hyperparameters fixed for initial 10 iterations and updated them every step thereafter. The overall procedure of bNMF inference is summarized as follows:

1. Choose a maximum rank $r_{max}$ and consider all rank $r = 2, \cdots, r_{max}$. For each $r$,

   a. Factorize count matrix $X$ using a random initial guess for $W^{(p)}$ and $H^{(p)}$ sampled from Equation (9) (see Algorithm 1 in Cemgil (2009)). Store the corresponding log evidence $U_p(r)$.
   b. Repeat **a** for a given number of different initial conditions and find $p^* = \arg\max_p U_p(r)$. Store $W^{(p^*)}$ and $H^{(p^*)}$ for the rank $r$.
2. Construct the evidence versus rank profile via $\{U_{p^*}(r)\}, r = 2, \cdots, r_{max}$. Find the optimal rank $r_{opt}$ for which $U_{p^*}(r)$ is maximum (Fig 1B; see below).
3. Construct the subgroup tree connecting rank $r = 2$ and $r_{max}$ (see below).
4. Use $(W, H)$ under rank $r_{opt}$ to derive metagene lists and assign cells to subgroups (Fig 1C).

The computational requirements of bNMF inference scaled linearly with increasing matrix dimensions (Fig S11). Because factorizations for each rank and initial conditions are independent, computation is easily distributed into multiple cores with linear speed-up.

### Determination of optimal rank

We determined the heterogeneity class and optimal rank based on evidence defined by Equation (8). We assumed that the support from data for rank $r'$ is statistically more significant compared to rank $r$ if the Bayes factor satisfies:

$$\text{BF} = \frac{\Pr(X|\Theta^*, r')}{\Pr(X|\Theta^*, r)} > T^n, \quad (10)$$

where $T$ is a threshold (Held & Ott, 2018). The exponent $n$ takes into account the fact that data $X$ contains $n$ samples. We used $T = 3$ in this work. In terms of the log evidence per matrix element $\epsilon(r) = [\ln \Pr(X|\Theta^*, r)]/nm$, we then have:

$$\epsilon(r') - \epsilon(r) > L = (\ln T)/m. \quad (11)$$

The left-hand-side of Equation (11) becomes the slope of log evidence if $r' = r + 1$. We used the following procedure to classify heterogeneity type and determine the optimal rank:

1. Replace evidence profile data $\epsilon(r)$ for $r = r_{min}, \cdots, r_{max}$ by its cubic-smoothing splined points to reduce artefacts from statistical noise. We used "smooth.spline" function in R with degrees of freedom d.f. $= \min(10, r_{max} - r_{min} + 1)$. We used a larger d.f. if fit was inadequate. Find $r^* = \arg\max_r \epsilon(r)$.
2. If $|\epsilon(r_{max}) - \epsilon(r^*)| > L$, the class is type I and $r_{opt} = r^*$.

3.  Otherwise, the class is type II. Compute the slope:

$$s(r) = \begin{cases} \epsilon(r+1) - \epsilon(r) & \text{if } r = r_{\min}, \\ \epsilon(r) - \epsilon(r-1) & \text{if } r = r_{\max}, \\ [\epsilon(r+1) - \epsilon(r-1)]/2 & \text{otherwise,} \end{cases} \quad (12)$$

and $r_{\mathrm{opt}}$ is the lowest rank for which $s(r) < L$. If no such rank exists, $r_{\mathrm{opt}} = r_{\max}$.

## Software availability

An R package implementing the algorithm is available as a Bio-conductor package, https://bioconductor.org/packages/ccfindR.

## Simulated data

We generated simulated data to characterize rank determination of bNMF algorithms in two different ways. First, for given numbers of genes $m$, rank $r$, and the total number of cells $n = rn_c$ ($n_c = 20$, $r = 10$ in Fig 2, such that $n = rn_c = 200$), we set the coefficient matrix $H$ such that $H_{kj} = 1$ for $j = (k-1)n_c + 1, \cdots, kn_c$, $k = 1, \cdots, r$, and zero otherwise. The basis matrix $W$ was set by dividing $m$ rows into $r$ groups and assigning elements of each group of rows by sampling from multinomial distributions of given total counts with uniform probabilities. The count matrix $X = WH$ was used after randomly shuffling rows and columns. ML-NMF and bNMF inferences used 50 different initial conditions for each rank. PCA-based analysis (Fig 2D) used Seurat (Macosko et al, 2015) using 10 principal components (Fig S1C). We varied the resolution parameter, as an input to "FindCluster" function, with default values of other parameters. The bNMF inference was repeated for different matrix sizes as indicated in Fig 2C. We used a realization of simulated data generated under rank 5 to determine the distribution of relative outlier cells (Fig S2). The bNMF factorization results were visualized using tSNE (Fig S2B) and relative outliers were identified using the function "cov.mcd" in the R package "MASS" with default parameters.

We tested the convergence of bNMF by generating a second set of simulated data using basis $W$ and coefficient $H$ matrices, whose elements were sampled from their $\gamma$ prior distributions with a given set of hyperparameters. We chose these hyperparameter values in Fig S3 as $a_w = a_h = 0.1$ and $b_w = b_h = 1$. The number of features ("genes") was fixed as 100, and we considered three values for the total number of cells ($n = 10$, 100, and 1,000). We computed the product of sampled matrices $W$ and $H$, whose elements were used as the mean values for the Poisson counts. Multiple realizations (100) of these count matrices for the single set of mean values given by $WH$ were generated for each sample size, and bNMF inference was performed separately (10 different initial conditions per rank) to determine the log evidence versus rank profiles, optimal rank statistics, and the distribution of final hyperparameter values (Fig S3).

## Gene selection

We applied quality control filtering to count matrix and gene/cell annotation data to select features with high variance for subgrouping (Fig S4). We used processed RNA count matrices of publicly available single-cell data sets (Table S1). We computed the variance to mean ratio (VMR) for all genes and selected genes with VMR above a cutoff. We also used a cutoff for the number of cells expressing each gene such that only those genes with nonzero counts in a minimum number of cells would be included. For a subset of samples, we further expanded the pool of genes such that those with relatively lower variance but with potentially nontrivial count distributions would also be included: for each gene filtered out by the criteria above, we constructed its count distribution histogram, which is typically peaked at zero count and mono-tonically decreases with increasing count. For a varying fraction of genes, this histogram contained a mode (a local maximum at a nonzero count). We moved filtered genes back into the selection when such a mode existed in its count distribution (Fig S4). Data sets with unique molecular identifier counts were used without normalization. For data sets reporting transcripts per million or fragments per kilobase per million, we took log-transformed levels of these quantities as pseudo-Poisson counts.

## Gold standard and PBMC data sets

We used six publicly available data sets (Yan et al, 2013; Biase et al, 2014; Deng et al, 2014; Pollen et al, 2014; Kolodziejczyk et al, 2015; Goolam et al, 2016) previously used in benchmarking SC3 (Kiselev et al, 2017) and SIMLR (Wang et al, 2017). The accession numbers of these data sets were GSE57249, GSE36552, E-MTAB-3321, GSE45719, SRP041736, and E-MTAB-2600. Pollen data set was downloaded from https://hemberg-lab.github.io/scRNA.seq.datasets/human/tissues/. We used VMR-based and cell-count–based gene filtering to obtained processed count matrices of dimensions shown in Table S1.

We used fresh PBMC and purified blood cell data (Zheng et al, 2017) from https://support.10xgenomics.com/single-cell-gene-expression/datasets. We generated two samples with different sizes by down-sampling original PBMC data set ($n = 34,289$ and $n = 6,857$; 11,212 genes). We applied ML-NMF and bNMF (Fig 2E–G) to the smaller data set, finding the solution with ML (ML-NMF) and evidence (bNMF). To annotate each cluster (Fig 3B and C), we first computed correlations between the mean RNA counts of bNMF subgroups and purified blood cell groups. We then used the "solve-LSAP" function of the R package "clue" (Hornik, 2005) to find the most likely assignment of bNMF subgroups to purified cell types. The annotation shown in Fig 3B is a consensus of this assignment and the metagene/marker lists (Fig 3A).

With Seurat, we used the smaller PBMC data set ($n = 6,857$) and applied the quality control procedure of cell filtering with the proportion of mitochondrial genes less than 0.08 and minimum unique molecular identifier count of 100. Variable genes were selected with the range of mean expression level between 0.02 and 3 and log VMR above 0.5, which yielded 1,773 genes and 6,847 cells. We used seven principal components based on the elbow plot (Fig S1F) and varied the resolution parameter to obtain Fig 2H.

We assessed the reliability of cell-type identification by bNMF using random mixtures of purified blood cell data containing from two to seven cell types (Fig 4A–F). Hundred random realizations of up to seven cell types (CD8$^+$ CTLs, CD19$^+$ B cells, CD14$^+$ monocytes,

CD4$^+$ Th, Treg, CD56$^+$ NK cells, and CD34$^+$ HSCs, each containing 100 cells, respectively) were generated by sampling columns from the purified cell count matrices and the count matrices of each realization were constructed by combining these columns. Rank determination and metagene identification in bNMF were performed for each realization after selecting genes with minimum VMR ratio of 1 and minimum number of 10 cells expressing the gene. Factorizations were performed for 50 different realizations of mixture, each with 10 initial conditions. Rank values with maximum evidence from each realization were extracted to obtain distributions shown in Fig 4A–F. Annotation scores in Fig 4I were calculated for four-cell-type mixtures first for the case of equal composition of Fig 4C and then for the "common + rare" mixtures containing 180, 20, 20, and 180 cells of CTLs, B cells, monocytes, and Th cells, respectively. Comparison of cell-type composition prediction from single-cell analysis and bulk data deconvolution was performed by summing RNA counts of fresh PBMC (Fig 3D) and a realization of seven-blood-cell mixtures (CTLs, B cells, monocytes, CD4$^+$ Th, Treg, NK cells, and HSCs of count $n$ = 100, 80,,120,100, 80, 80, 80, respectively; Fig 3E) for all genes under consideration. We used these bulk counts as input to CIBERSORT at https://cibersort.stanford.edu/ with default parameters.

### Metagene identification

To characterize subgroups derived from bNMF inference under the optimal rank $r$ (9 for PBMC), we took the basis matrix elements $W_{ik}$ and analyzed them column by column. For each subgroup indexed by $k$ = 1, …, $r$, we rescaled the vector $W_{ik}$ by dividing each row (the basis component of gene $i$ in each subgroup $k$) by its geometric mean over $k$, such that different genes would have basis components that are comparable in magnitude. For $k$ running from one to $r$, we then reordered the rows of $W$ such that the $k$-th column would have monotonically decreasing magnitudes from top to bottom. We subsequently looked at the top $m$ rows of the sorted matrix and selected genes whose rows within the submatrix given by $i$ = 1, …, $m$ had their maximum elements at position $k$. The genes corresponding to these rows were defined as the metagenes of the subgroup $k$. This definition avoids picking genes that feature strongly in one subgroup but even more so in other subgroups, instead focusing on those that help identify the given subgroup uniquely (Carmona-Saez et al, 2006). These steps were repeated for all $k$. Note that the maximum number of metagenes per subgroup is $m$ and we often found the actual numbers to be smaller. Marker genes, preselected for PBMC in addition to the genes with high variance, were considered together with $m$ genes in the above procedure, such that the actual maximum size of the metagene-plus-marker set was $m$ plus the total number of markers. As can be seen in Fig 3A, however, each marker gene appears only once in the subgroup in which the marker contribution is strongest.

### Subgroup tree construction

We inferred hierarchical relationships between subgroups obtained under different ranks by comparing cellular subgroup memberships of neighboring ranks. Specifically, we used the series of coefficient matrices with elements $H_{kj}^{(r)}$ for rank $r$ = 2, ⋯, $r_{opt}$,

where $r_{opt}$ is the optimal rank, to derive the subgroup index of cell $j$ under rank $r$ given by $c_{j,r}$ = argmax$_k H_{kj}^{(r)}$. For each subgroup $k$ under rank $r$ + 1, we then tabulated the subgroup index $c_{j,r}$ of all cells $j$ belonging to subgroup $k$ and defined the subgroup of origin by:

$$l_{k,r+1} = \arg\max k' \sum_{j \in k} \delta(k', c_{j,r}), \tag{13}$$

where $\delta(x, y)$ = 1 if $x = y$ and zero otherwise and the summation is over all cells belonging to subgroup $k$ under rank $r$ + 1. The subgroup of origin $l_{k,r+1}$ is the subgroup under rank $r$ with the highest count of cells in the subgroup $k$ under rank $r$ + 1. In rare cases where there were ties in ranking for the subgroup of origin count, we randomly broke the tie such that $l_{k,r+1}$ would be uniquely defined for all $k$. We then grew the tree at a given $r$ by connecting the subgroup $k$ under rank $r$ + 1 to the subgroup $l_{k,r+1}$ under rank $r$. In most cases, this step resulted in bifurcation of a subgroup under rank $r$, but triple-branching also occurred occasionally. We repeated this procedure sequentially for $r$ = 2, ⋯, $r_{opt}$ − 1 to complete the tree.

### Pancreatic tissue sample

We downloaded human pancreatic tissue single-cell count matrix (patient 1; Baron et al, 2016) via accession number GSE84133. We used all 1,937 cells in the count matrix and selected 2,454 genes using minimum VMR of 2 and minimum number of 100 cells expressing each gene. Rank scan for $r$ up to 40 used 20 initial conditions for each rank.

### Cancer samples

We used processed RNA count matrices of cancer samples via accession numbers GSE81383, GSE97168, GSE70630, GSE89567, GSE72056, GSE114724, GSE117156, and GSE103322, for melanoma cell culture, lung cancer immune cells, oligodendroglioma, astrocytoma, melanoma, breast cancer immune cells, MM, and HNSCC, respectively (Table S1). We used all cells and selected genes using thresholds for VMR and number of cells expressed as indicated in Fig S4 to obtain count matrices of dimensions shown in Table S1. For MM samples, immunoglobulin genes were excluded (Ledergor et al, 2018) in addition to VMR-based filtering. We chose patient ID BC09 (tumor 01) for the breast cancer immune cell sample (Azizi et al, 2018). For MM samples, we used patient IDs MGUS01, SMM01, and MM01 (Ledergor et al, 2018).

# Supplementary Information

## Author Contributions

J Wang: conceptualization, data curation, formal analysis, supervision, investigation, methodology, project administration, and writing—original draft, review, and editing.

J Woo: data curation, software, formal analysis, visualization, methodology, and writing—original draft, review, and editing.

B Winterhoff: resources, data curation, investigation, and writing—review and editing.

T Starr: resources, data curation, investigation, and writing—review and editing.

C Aliferis: conceptualization, resources, data curation, investigation, and writing—review and editing.

## Conflict of Interest Statement

The authors declare that they have no conflict of interest.

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
