## [Reviewer comments · Life Science Alliance]

De novo prediction of cell-type complexity in single-cell RNA-seq and tumor microenvironments

Jun Woo, Boris Winterhoff, Tim Starr, Constantin Aliferis, and Jinhua Wang
DOI: 10.26508/lsa.201900443

Review timeline:

Submission Date:	29 May 2019
Editorial Decision:	29 May 2019
Revision Received:	24 June 2019
Accepted:	24 June 2019

Report:

(Note: Letters and reports are not edited. The original formatting of letters and referee reports may not be reflected in this compilation.)

Please note that the manuscript was previously reviewed at another journal and the reports were taken into account in the decision-making process at Life Science Alliance. Since the original reviews are not subject to Life Science Alliance's transparent review process policy, the reports and author response cannot be published.

1st Editorial Decision

29 May 2019

Thank you for transferring your revised manuscript entitled "De novo prediction of cell-subtype complexity in single-cell RNA-seq and tumor heterogeneity" to Life Science Alliance. Your work was reviewed at another journal twice before, and the reviewer reports and your responses have been transferred to us with your permission.

Two reviewers criticized the lack of broader conceptual advance provided by your study. This is not a concern for publication here and we appreciate how you addressed the individual concerns of all reviewers, including now a more thorough discussion of existing approaches for choosing k in single-cell RNA-seq analyses and benchmarking your approach. We would thus be happy to publish your paper in Life Science Alliance pending final revisions necessary to meet our formatting guidelines:

- please upload all figures (also supplementary figures) as individual files, the legends should only be in the main manuscript docx file
- please make sure to add for each author the contributions in our submission system
- please link your ORCID iD to your profile in our submission system, you should have received an email with instructions on how to do so

To upload the final version of your manuscript, please log in to your account with the login name (you will have to request a new password):

A. FINAL FILES:

B. MANUSCRIPT ORGANIZATION AND FORMATTING:

2nd Editorial Decision

24 June 2019

Thank you for submitting your Methods entitled "De novo prediction of cell-type complexity in single-cell RNA-seq and tumor microenvironments." It is a pleasure to let you know that your manuscript is now accepted for publication in Life Science Alliance. Congratulations on this interesting work.

DISTRIBUTION OF MATERIALS:

Again, congratulations on a very nice paper. I hope you found the review process to be constructive and are pleased with how the manuscript was handled editorially. We look forward to future exciting submissions from your lab.